# Degradation of Veterinary Antibiotics in Swine Manure via Anaerobic Digestion

**DOI:** 10.3390/bioengineering7040123

**Published:** 2020-10-09

**Authors:** Ali Hosseini Taleghani, Teng-Teeh Lim, Chung-Ho Lin, Aaron C. Ericsson, Phuc H. Vo

**Affiliations:** 1Division of Food Systems and Bioengineering, University of Missouri, Columbia, MO 65211, USA; ahbq6@mail.missouri.edu; 2School of Natural Resources, University of Missouri, Columbia, MO 65211, USA; LinChu@missouri.edu (C.-H.L.); phucvh2410@gmail.com (P.H.V.); 3Department of Veterinary Pathobiology, University of Missouri, Columbia, MO 65201, USA; ericssona@missouri.edu

**Keywords:** anaerobic digester, antibiotics removal, antimicrobial, biogas, chlortetracycline, Tylosin

## Abstract

Antibiotic-resistant microorganisms are drawing a lot of attention due to their severe and irreversible consequences on human health. The animal industry is considered responsible in part because of the enormous volume of antibiotics used annually. In the current research, veterinary antibiotic (VA) degradation, finding the threshold of removal and recognizing the joint effects of chlortetracycline (CTC) and Tylosin combination on the digestion process were studied. Laboratory scale anaerobic digesters were utilized to investigate potential mitigation of VA in swine manure. The digesters had a working volume of 1.38 L (in 1.89-L glass jar), with a hydraulic retention time (HRT) of 21 days and a loading rate of 1.0 g-VS L^−1^ d^−1^. Digesters were kept at 39 ± 2 °C in incubators and loaded every two days, produced biogas every 4 days and digester pH were measured weekly. The anaerobic digestion (AD) process was allowed 1.5 to 2 HRT to stabilize before adding the VAs. Tests were conducted to compare the effects of VAs onto manure nutrients, volatile solid removal, VA degradation, and biogas production. Concentrations of VA added to the manure samples were 263 to 298 mg/L of CTC, and 88 to 263 mg/L of Tylosin, respectively. Analysis of VA concentrations before and after the AD process was conducted to determine the VA degradation. Additional tests were also conducted to confirm the degradation of both VAs dissolved in water under room temperature and digester temperature. Some fluctuations of biogas production and operating variables were observed because of the VA addition. All CTC was found degraded even only after 6 days of storage in water solution; thus, there was no baseline to estimate the effects of AD. As for Tylosin, 100% degradation was observed due to the AD (removal was 100%, compared with 24–40% degradation observed in the 12-day water solution storage). Besides, complete Tylosin degradation was also observed in the digestate samples treated with a mixture of the two VAs. Lastly, amplicon sequencing was performed on each group by using the 50 most variable operational taxonomic units (OTUs)s and perfect discriminations were detected between groups. The effect of administration period and dosage of VAs on Phyla *Firmicutes* Proteobacteria, Synergistetes and Phylum Bacteroides was investigated. These biomarkers’ abundance can be employed to predict the sample’s treatment group.

## 1. Introduction

Nowadays, a wide variety of antibiotics are being used in animal farms to cure, prevent and also to improve the growth of animals, accounting for more than 52% of total antibiotics consumption in the world [1,2,3]. Due to the rapid effect of antibiotics and low cost of them, daily use of it rocketed during the last two decades) [4]. Although in 2017, attempts to restrict using antibiotics took place, from 2009 to 2016 use of several veterinary antibiotics (VAs) was raised by 36.8% on average [4]. Many of these compounds have weak absorption within the animal gut and intestine during digestion, resulting in the excretion of potent parent and daughter products [5]. A high percentage of the antibiotics (60–90%) is excreted without metabolism in urine and feces, leading to potential human and ecological health risks for soil and water [6,7,8,9]. In addition, based on a study of Alexy et al. [10], most of the antibiotics being used are not biodegradable, with degradation extents varying between 4 and 27%.

Since a vast volume of manure is being produced each year due to concentrated animal feeding operations (CAFOs) and is mostly being applied to solid materials as a fertilizer, the long-term presence of such antibiotics in manure with even trace concentrations (i.e., ng/L) could lead to the formation of antibiotic resistance genes (ARGs) [11]. The microbacterial resistance will result in higher medical costs, longer treatment periods, and increased mortality [12,13]. Animal farms typically utilize simple treatment systems, which are mainly AD, stockpiling, composting, wetlands, or lagoons ([3,14]). These simple treatments might not be sufficient to prevent the appearance of ARGs [15]. There are new approaches to remove or reduce antibiotics, including activated carbon adsorption, membrane filtration, advanced oxidation processes ([14,16,17,18,19]), however, they require advanced technical supervision as well as extreme expenses. Furthermore, the VAs could pollute the soil and water, then the human food chain through crops and animal-derived foods [20,21,22,23,24]. Moreover, the residue of VAs in the AD process could sustain microbes under the minimum inhibitory concentration (MIC), fostering the selection for ARGs by microbes [25,26].

Anaerobic digesters can produce biogas as well as removing VAs and ARGs [27,28,29]. In contrast, composting [30] requires less monitoring, has a more stable working system and a generally higher removal rate for certain VAs, but consumes energy and occasionally has less capability to remove ARGs than AD. Xie et al. [31] concluded that thermophilic composting of cow manure would result in ARG mitigation, lowering 16S rRNA with tetracycline, sulfonamide and fluoroquinolone resistance genes, however, not effective with aadA, aadA2, qacED1, tetL, cintI1, intI1, and tnpA04. A similar study of dairy manure composts showed satisfactory treatment of antibiotic-resistant E. coli and Salmonella, yet some antibiotic-resistant Enterobacter spp. and multidrug-resistant Pseudomonas spp. population raised after application of these composts to rangeland soils in Texas [29]. Prado et al. [32] aimed to use an aerobic reactor with activated sludge to track the fate of tetracycline (TC) and Tylosin as antibiotics. Both TC and Tylosin were not biodegradable in this type of reactor. Research also determined that the biosorption of both antibiotics appeared to be most favorable for TC.

Joy et al. [33] investigated the behavior of three antibiotics (bacitracin, chlortetracycline, and tylosin) and two classes of ARGs (Tet and Erm), which were monitored in swine manure slurry under anaerobic conditions. First-order decay rates were determined for each antibiotic with half-lives ranging from 1 day (chlortetracycline) to 10 days (tylosin). Angenent and Wrenn [34] examined the effects of an anaerobic sequencing batch reactor (ASBR) on the removal of antibiotic tylosin. They observed no inhibitory effect on biogas production, but some macrolide–lincosamide– streptogramin B (MLSB)-resistant bacteria appeared. Shi et al. [35], discovered that a certain dosage of tetracycline (TC) and sulfamethoxydiazine (SMD) could reduce biogas production. They also noticed the rapid disappearance of antibiotics (more than 50%) in the first 12 h. However, they were not sure about whether it was being degraded or just absorbed into solid materials. A similar study was conducted by Beneragama et al. [36], who confirmed the efficiency of AD of antibiotics in dairy manure. They also utilized thermophilic microorganisms (working in 55 °C or 131 °F). Results showed no inhibition in gas production and the efficiency of the reactor.

Approximately 80% of the 16,000 metric tons of antibiotics sold annually in the U.S. are used in animal husbandry [37]. These antibiotics can be transported to run-off water, groundwater, soil, and finally, plants [38,39,40,41,42,43]. In 2030, antibiotic resistance would cost USD 3.4 trillion due to subsequent mortality and substitute treatment [44]. Therefore, according to this extensive use of antibiotics and their stability in the environment, seeking an efficient and financially feasible method is vital. Besides, limited findings are available about the threshold concentrations of different classes of antibiotics in manure that can be removed during the AD process and the interaction between anaerobic digesters and antibiotics.

The objectives of this study were to evaluate (i) anaerobic digestion efficiency on the removal of chlortetracycline (CTC) and Tylosin, (ii) inhibitory behavior of VAs on the reactors, and (iii) the effect of these antibiotics on microbial dynamics in anaerobic digesters with swine manure. The novelty of this research is the imitation of on-farm mesophilic anaerobic digesters that were loaded frequently with manure from a commercial pig farm, while operating over several months. The research focused on widely used antibiotics and emphasized proper concentration and duration at which the antibiotics were administered (following manufacturer’s recommendation and average pig weights), and amount of antibiotics excreted by the animals based on a literature review. Critical operating variables of the digesters, including pH and biogas productions were monitored closely, similar to what most on-farm AD technicians are employing to monitor AD performance, which require no sophisticated analytical expertise. The bench-scale anaerobic digesters were relatively larger and semi-continuously loaded for over several months, while many of the previous studies only focused on inhibition effect and usually using batch reactors, which are different from actual on-farm AD conditions.

## 2. Methods

The current study focuses on utilizing an AD process to assist in removing VAs and finding the efficiency and practicality of the reactors. Antibiotics are chosen to be spiked meticulously, based on their importance, usage in feedstock and their danger to the environment. Besides, the dosage of antibiotics was close to concentrations administrated for animals, absorbed, and then excreted, to imitate the real condition. Our reactors are fed with swine manure, which has been tested for background concentration of VAs, to diminish the chance of interference. CTC was injected into anaerobic sequencing batch reactors (ASBR) with doses of 263, 280, and 298 mg/L for each spike and a total of three injections every two days while Tylosin doses were 88, 175, and 263 mg/L for each injection and a total of 5 injections every two days.

Several factors are contributing to reactor performance and biogas production. Temperature is one of them. Different types of bacteria work on various temperature ranges and some of them are highly susceptible to temperature fluctuation. Besides, the pH and alkalinity of the environment in which bacteria are growing should be near neutral and consistent. Therefore, the temperature was kept at around 39 °C (102 °F). In addition, we were recording incubator temperature and humidity for tracking the performance of our incubator. The methane forming bacteria are very sensitive to slight changes in organic loading, pH, and temperature (a temperature change greater than 2 degrees of Fahrenheit per day will affect the methane formers).

### 2.1. Feedstock for Anaerobic Digestion

Manure samples were collected from a mid-central Missouri commercial swine farm. The farm was VA free for the finishing pigs, located in Versailles, Missouri, USA. Furthermore, to make sure that no antibiotics existed in the solid manure used, it was analyzed to eradicate any interference or error. Because the farm has shallow pits, the manure would be less than one month old. After collecting manure, buckets full of manure were kept frozen at −20 °C (−4 °F) until they were used as feedstock for the reactors. Once manure was needed, one of these big buckets was thawed down and separated into a small bucket (usually 4 L (L) in volume). Just one of these small buckets was in the refrigerator for feeding; the rest were kept in a freezer to keep it unchanged as much as possible. Total and volatile solids (TS and VS) of each big bucket were tested to evaluate the proper feeding ratio. The total solid (TS) of the solid manure was 25.89%, and the volatile solid (VS) was 82.02% of the TS. There was no test conducted to verify the potential effect of the freezing, although there were little observed changes in biogas production between refrigerated and frozen manure in the last year of AD tests.

The inoculum was collected from semi-continuous AD jars of previous tests (Wang et al. [45]), which were steadily producing biogas for over three months, and the feedstock was swine manure with organic loading rate (OLR) of 1 g-VS L^−1^ d^−1^ only. The total solid (TS) of inoculum was 2.20%, and the volatile solid (VS) was 64.92% of TS.

### 2.2. Experimental Design

Tests were carried out with laboratory size jars as reactors (adjusted for AD). Antibiotics were added to reactors with different concentrations of CTC and Tylosin to monitor antibiotic removal and gas production variations in those mesophilic reactors. Nine laboratory-scale jars as anaerobic bioreactors with the working volume of 1.375 L were kept at 39 ± 2 °C (102 °F to 105 °F) in the incubator. The jars are being fed with VA-free swine manure at 1g-VS per L-day, with 21 days hydraulic retention time (HRT). The volume of the feed given every two days is measured based on HRT and our reactor volume. Because our HRT is 21 days and the reactor volume is 1.375 L, so 0.131 L of our reactor liquid were removed and replace by feedstock (the digesters were fed every two days) [46].

Each jar was connected to 10-L Tedlar bags to collect produced biogas, and the volume was measured every four days [47]. A custom-built device was used to help distribute the biogas evenly in the bag, so the height of the bag could be measured more accurately. By utilizing a predetermined model, the volume of each bag was then estimated by bag height. Besides, to prevent any leakage of the Tedlar bags, each time two of the bags were randomly tested for possible leakage before emptying. Additionally, tubes, caps, and any connective parts were tested for leakage. After biogas measurements, bags were emptied safely and burned.

The experiment consisted of nine jars; three of them were spiked with CTC, three with Tylosin and the last three with both CTC and Tylosin, to observe the combined effect or any interaction between two types of antibiotics (Figure 1). Furthermore, to investigate the efficiency of the AD process, we added six more jars, filled with distilled water and the headspace with N_2_. Three of these jars were being kept in the incubator at the same temperature of the digester jars (39 °C), the rest were being kept in the room temperature to monitor the effect of the temperature. The same pattern of antibiotics concentration was conducted for control jars, two groups of three jars. Retention time and sampling procedures were identical.

### 2.3. Antibiotics

The two most widely used antibiotics were selected based on consumption rate and market share of different antibiotic classes in the United States [4]. Antibiotics used in this experiment were CTC as chlortetracycline HCI and tylosin as tylosin tartrate. Commercial grade CTC was bought from “PharmGate Animal Health; Omaha, Nebraska” with the brand of “Pennchlor 64”. Commercial grade tylosin used was from “Elanco Animal Health; Indianapolis, Indiana” with the brand of “Elanco”. Moreover, to prepare standard samples for LCMS/MS, both antibiotics were ordered as the analytical grade from “Sigmaaldrich”, St. Louis, Missouri. Chlortetracycline hydrochloride, VETRANAL™, analytical standard, with CAS number of 64-72-2 and tylosin, United States Pharmacopeia (USP) Reference Standard, with CAS number of 1401-69-0 were used for standard solutions.

We followed the prescription on the labels to imitate the real condition in a barn. Consequently, the recommended dosage for tylosin was 66 ppm in drinking water. For Swine Dysentery, adding tylosin to drinking water should be continued for 3 to 10 days, depending upon the severity of the infection. For CTC, the recommended dosage was 22 mg/kg body weights per day. The duration of treatment is 3 to 5 days depending on the infection. Pigs are generating 4.28 L of manure per day on average. Additionally, it is assumed that the average body weight of a pig is around 68 kg (finishing pigs weigh around 45–113 kg). These assumptions would help us estimate the concentration we should inject in our reactors, by considering the excretion rate and metabolism percentage.

The stabilization time for the reactors and the microbial community was expected to be one to three months, until the biogas production, digestate pH, and alkalinity trend became flat. For the current research, the digester was fed for eight weeks or 2.5 times the retention time. Important operating variables, including organic loading rate based on total volatile solids (TVS), solid content, temperature, mixing (swirling the jar daily), and foaming (if any) were recorded. Digester alkalinity and pH were monitored weekly by measuring the digestate.

For Lower range concentration, the lowest factor in each section was used. For instance, to calculate the lower band of tylosin, 11.35 L per day as pig’s drinking volume, 50% excretion level applied and 4.28-L excretion per day was selected. For upper range concentration, the highest factor in each section was used. For instance, to calculate the upper band of tylosin, 18.93 L per day as pig’s drinking volume, 90% excretion level and 4.28-L excretion per day was selected. The average concentration is the average of the lower and upper concentration. Recalling that jars were loaded every two days with a mixture of solid manure and water, VAs added with feed had a concentration of day 1 plus day 2.

Since antibiotics are being added to the water part of the feeding (not to solid part), therefore solubility of the VAs should be checked. Table 1 is a summary of the solubility of CTC and tylosin in the water at 20 °C:

Considering the solubility of CTC and tylosin in the water at 20 °C, there was no problem with CTC and tylosin solving limit individually. However, one set of three jars was used, which we decided to use to test the combined effect of antibiotics, so we had to mix two antibiotics in the same volume of water (0.103 L). There is always a chance of interference between two types of chemicals, especially when they are being added near their solubility limit. Thus, the decision was made to add CTC directly to the water, transfer it to the reactor and then add tylosin powder separately to the reactor. Other solvents such as methanol or ACN were dismissed because of their adverse effect and interference with the reactor’s performance (an independent test was conducted to evaluate the impact of adding methanol onto AD performance; details are not included in this paper). Table 2 summarizes the recommended VA concentrations based on the manufacturer’s recommendation and corresponding dosages considering the ranges of dosage, water consumption, and excretion rate.

As previously mentioned, we recorded biogas production for at least two HRTs, before and after introducing the antibiotics. Table 3 illustrates the added VAs concentration in each reactor. As shown below, the first group is being administrated only with CTC, the second group with both CTC and tylosin and the last, with only tylosin.

The pH of the digestate was measured every two days while adding antibiotic, with pH meter (PINPOINT, American Marine Inc., Ridgefield, CT, USA). Using pH data, microbial activity of the digester and the reactor performance is projected. However, pH can also be affected by alkalinity. For quality assurance, alkalinity tests were also carried out.

The CO_2_ concentration of the biogas was measured with a standard combustion analyzer (Bacharach Fyrite Classic Combustion Analyzer, New Kensington, PA, USA) every eight days. The concentration of CO_2_ was measured every four days during the antibiotic addition period. Comparative tests using a gas spectrometry were used to check how accurate our measurements were. Below is a comparative table that illustrates accuracy control values (Table 4). The gas chromatograph device was (GC-2014, Shimadzu, US) with a thermal conductivity detector (TCD) using a ShinCarbon ST 80/100 Column (Restek, US) [48].

Because the administration times for CTC and tylosin were different (6 days for CTC and 10 days for tylosin), jars with CTC spikes were sampled at the end of day 6, while tylosin-spiked jars were sampled at day 10. For jars with the combined CTC and tylosin, samples were taken at both day 6 and day 10. Since VAs were added every two days with feed, the sampling would occur two days after the last spike. Samples were frozen at −20 °C immediately. Gas production, pH and CO_2_ level were considered the vital data, which were recorded before, during and past spikes.

## 3. Extraction and Chemical Analysis

### 3.1. Sample Preparation

Two grams of the sample were transferred to 50-mL “Corning™ PP Centrifuge Tubes (polypropylene) and 5 mL of phosphate buffer (0.14M) was added. Following the pH adjustment, 200 µL of internal standard (Sulfamethazine phenyl-13C6) was fortified and the antibiotics were extracted with 25 mL of acetonitrile (ACN) with sonication for an hour. Following the sonication, the samples were centrifuged for 15 min in 4000rpm, at 4 °C (39.2 °F) with a Sorvall LYNX 6000 Superspeed Centrifuge (Thermo Scientific™, Waltham, MA, USA), and the supernatant was collected. The same extraction process was repeated with 15 mL ACN; both supernatants were combined. Twenty-milliliters of the extract was transferred to the test tube, and the solvent was further evaporated to 2 mL under a stream of nitrogen gas. The extract was diluted with 18 mL of DI water before solid-phase extraction (SPE).

### 3.2. Solid-Phase Extraction

The antibiotics were extracted by a Waters Oasis-HLB SPE cartridge (Oasis HLB 12 cc Vac Cartridge, 500 mg Sorbent per Cartridge, 60 µm Particle Size). The solid-phase extraction cartridges were preconditioned in an order with 10 mL ACN, 10 mL DI water all with the rate of (2 mL/min). The sample was subsequently introduced to the cartridge at a flow rate of 2 mL/min. The impurity in the cartridge was washed by using 10 mL DI water for 5 min of vacuum drying. The antibiotics retained on the cartridges were eluted with 8 mL of methanol followed by 8 mL of ACN with a flow rate of 2 mL/min. The eluate was evaporated by a gentle stream of nitrogen at 15 L/min in a water bath at 35 °C and concentrated to 10–20 µL. The extract was further filtered via a 0.22-µm Anotop inorganic filter (Sigma Aldrich) and was ready for antibiotic analysis [49].

### 3.3. LC-MS/MS Analysis

The concentrations of antibiotics were determined by a Waters Alliance 2695 High-Performance Liquid Chromatography (LC-MS/MS) system coupled with Waters Acquity TQ triple quadrupole mass spectrometer (MS/MS). The analytes were separated by a Phenomenex (Torrance, CA) Kinetex C18 (100 mm × 4.6 mm; 2.6 µm particle size) reverse-phase column. The mobile phase consisted of 10 mM ammonium acetate and 0.1% formic acid in water (A) and 100% acetonitrile (B). The gradient conditions were 0–0.5 min, 2% B; 0.5–7 min, 2–80% B; 7.0–9.0 min, 80–98% B; 9.0–10.0 min, 2% B; 10.0–15.0 min, 2% B at a flow rate of 0.5 mL/min. The ion source in the MS/MS system was electrospray ionization (EI) operated in either positive (ES+) mode with a capillary voltage of 1.5 kV. The ionization sources were programmed at 150 °C and the desolvation temperature was programmed at 450 °C. The MS/MS system was in the multi-reaction monitoring (MRM) mode with the optimized collision energy. The ionization energy, MRM transition ions (precursor and product ions; Table 5), capillary and cone voltage, desolvation gas flow and collision energy were optimized by the Waters IntelliStart™ optimization software package [50]. The retention time, calibration equations, and limits of the detection for the analyses of metabolites are summarized in Table 5.

### 3.4. Statistical Analysis

The statistical analyses were carried out using a two-sample t-test with unequal variances from the statistical analysis (R Core Team, 2013) to compare biogas inhibition between groups and between different VAs concentrations. Significance was accepted at probabilities *p* ≤ 0.05 for all analysis. In addition, for amplicon sequencing, Bray–Curtis similarities and Jaccard similarities methods are used for this comparison. The Bray–Curtis dissimilarity is a method used to measure the structural variation between two different groups, based on counts at each group. Mathematically, the index of dissimilarity is:(1)$$\mathrm{BCij}=\frac{1-2\mathrm{Cij}}{\mathrm{Si}+\mathrm{Sj}}$$
where Cij is the sum of the lesser values for only those species on the intersection of two sets. Si and Sj are the total numbers of specimens at both sites. The range is between 0 and 1 [51].

The Jaccard similarity index compares members for two sets of data to quantify the resemblance between them, with a range from 0 to 1. The closer the number is to 1, the more similar the two populations are.
(2)$$J\left(A,B\right)=\frac{\left(A\cap B\right)}{\left(A\cup B\right)}$$

### 3.5. Sampling and DNA Isolation

Raw and digested manure samples have been analyzed by the MU Metagenomics Center for the microbial/taxonomy analysis using the 16S rRNA library sequencing methodology. The results show that over 60k sequences were identified, confirming that the taxonomy analysis of manure samples can be analyzed using the specific method.

In total, twelve samples were collected into 50-mL sterile centrifuge plastic tubes. The first three samples were taken from CTC-added jars, with low, medium, and high concentrations, sampled 6 days after the first addition of VAs. The next three were sampled from jars with the addition of a mixture of CTC and tylosin on day 6 and day 10. The last group, including samples 9 to 12 were taken from jars administrated only with tylosin and were sampled on day 10 of VAs addition. Prior to sampling, each jar was mixed thoroughly with a hand mixer for 1 min. During the time after sampling and before starting the amplicon sequencing, samples were frozen to prevent any interference with oxygen. According to the TissueLyser II (Qiagen, Venlo, The Netherlands), samples were incubated at 70 °C for 20 min with intervallic vortexing. Then, samples were centrifuged at 5000× *g* for five minutes at room temperature, and the supernatant conveyed to a 1.5-mL Eppendorf tube. Next, ammonium acetate was added, mixed, incubated on ice and centrifuged. The supernatant was then blended completely with a unit volume of chilled isopropanol and for 30 min incubated on ice. Products were then centrifuged at 16,000× *g* for 15 min at 4 °C. The supernatant was evaporated and removed; the DNA pellet was cleaned several times with 70% ethanol and resolved in 150 μL of Tris-EDTA. The rest of the method was performed, according to Ericsson et al. [52,53].

### 3.6. 16S rRNA Library Preparation and Sequencing

The DNA of extracted samples was tested at the University of Missouri DNA Core Facility. Bacterial 16S rDNA amplicons were created with a magnification of the V4 hypervariable region of the 16S rDNA gene with universal primers (U515F/806R) formerly established against the V4 region, edged by Illumina standard adapter sequences [54]. A single forward primer and reverse primers with a unique 12-base index were used in all reactions. PCR amplification was completed as follows: 98 °C(3:00) + (98 °C(0:15) + 50 °C(0:30) + 72 °C(0:30)) × 25 cycles + 72 °C(7:00) [52,53]. The amplified product from each reaction was mixed entirely; then purified and incubated at room temperature for 15 min. Products were washed with 80% ethanol several times and the dried pellet was resuspended in Qiagen EB Buffer (32.5 μL), incubated at room temperature for 2 min, and then placed on the magnetic stand for 5 min. The final amplicon pool was assessed using the Advanced Analytical Fragment Analyzer automated electrophoresis system, quantified with the Qubit fluorometer using the Quant-iT HS dsDNA reagent kit (Invitrogen), and diluted according to Illumina’s standard protocol for sequencing on the MiSeq [52].

### 3.7. Informatics Analysis

Constructing, data binning, and descriptive analysis of DNA sequences was performed at the MU Informatics Research Core Facility. FLASH software [55] was employed to group the contiguous sequences of DNA, and contigs were discarded if they turned out to be less than 31 after trimming for a base quality. Qiime v1.7 [56] software was used to carry out de novo and reference-based chimera detection and exclusion, and other contigs were allocated to operational taxonomic units (OTUs) with a significance of 97% nucleotide identity. Taxonomy was appointed to selected OTUs using BLAST [57] in comparison to the Greengenes database [58] of 16S rRNA sequences and taxonomy.

## 4. Results and Discussion

The presence of VAs in anaerobic digesters could have an inhibitory effect on biogas production [36,59,60], because VAs could disrupt microorganisms’ dynamics, especially when the concentration is high. Because AD is not efficient in degrading VAs completely, in the long term, AD reactors can also become a fostering environment for VAs that would help the development of new ARGs [61]. By scrutinizing the figures derived, some abnormalities were visible one week after the last spike, recalling that October 18 was the start date of the spiking antibiotics and final day was October 28 (Figure 2a). This biogas fluctuation started with a decline in samples spiked with tylosin and also a mixture of tylosin and CTC, immediately after the first spike. For CTC samples, this drop was delayed until early November. On November 11, it grew again and then reached its lowest point on November 23.

For the CTC plus tylosin, after a drop on October 22, and again on October 30, we witness a surge after that. Reactor #4 peaks on November 7 and reactor #6 peaks on November 19. Reactor #5 climbs steadily during this period. They start to drop in mid-November and reach their low at the end of November. Similarly, Figure 2b shows the same behavior, declining after the first spike until the end of October (last spike), followed by an upward trajectory. Likewise, this trend hits its bottom in early December.

Running a T-Test on biogas data implies that AD bacterial activity was immediately inhibited for samples that have tylosin in them (Figure 2b,c) (*p*-value = 0.005). Still, the bacteria either adapted or the inhibiting compound was removed from the system after a few weeks. Biogas production was untouched for CTC samples, yet for the mixture of CTC and tylosin, and tylosin alone, it was significantly lower, immediately after VA addition. The tylosin concentration in this experiment was 92 mg/L and less, complying with the findings of Mitchel et al. [62]. They concluded that the bioreactor containing 92 mg/L tylosin had less biogas for nearly 30 d until the system recovered. The biogas reduction for samples with tylosin and CTC was close to 14%, and for tylosin, samples was between 8 and 19%, with no dose-dependent relationship. On the other hand, Chelliapan et al. [63] found no biogas inhibition in an up-flow anaerobic stage reactor (UASR) containing 100–800 mg/L tylosin.

Erythromycin, another macrolide antibiotic caused 6–24% biogas reduction with 6–100 mg/L, and no dose-dependent relationship [64].

But CTC did not disturb the bacterial activity, substantiate the evidence that CTC antibiotic may present minimal AD biogas inhibition at concentrations less than approximately 70 mg/L occurring in the current study. Yin et al. [65] observed similar results; for a mesophilic anaerobic digester with the manure and CTC concentrations of 0, 20, 40, and 60 mg/kg. TS, no significant inhibition in biogas production occurred. Dreher et al. [66] showed that no inhibition of biogas production happened in anaerobic sequencing batch reactor with 28 mg/L CTC but that the volumetric composition of methane decreased by about 13–15%. Mixed results of the inhibition in the literature could be due to various reactor types, inoculum/manure ratio, inoculum and manure age and source, reactor size, and batch or continuous operation [67]. In this experiment, CTC concentration was probably lower than its required inhibitory level.

### 4.1. pH and CO_2_

pH value can demonstrate how well Acetogenesis and Methanogenesis bacteria are working. At the beginning of AD performance, Acetogenesis bacteria start to produce volatile acids that cause the pH to decrease. Subsequently, Methanogenesis bacteria convert the volatile acids to methane and CO_2_, and cause pH to increase. At HRTs with more than five days, the methane-forming bacteria begin to consume the volatile acids.

By comparing before and after the addition of VAs, it is evident that reactors are experiencing a fluctuating pH status (Figure 3a–c). The graph shows that variations immediately after antibiotic spike have increased intensively, with a rising trend. Following up, in the first week of November, almost all reactors reach their plateau. From then on, the gradual decline continued until November 17. Subsequently, reactors seemed to recover themselves with an increase in pH. By the end of November, pH returns to its average level of around 7.8.

The T-test on the pH data shows that pH values were significantly lower for the samples with tylosin in them (*p*-value = 0.05). However, CTC did not affect the pH significantly. Since fluctuations in pH level are not sharp, this indicates that VAs did not disturb the bacterial community substantially.

Nevertheless, none of the reactors became upset or affected intensively by the VA addition. pH fluctuation was ±0.16 maximum and it never dropped under 7.60. Similarly, the CO_2_ level has detectable alteration around the antibiotic spike date (Figure 4a–c).

Biogas produced is consisting of almost 50–75% of methane, 25–40% of carbon dioxide and other gases, depending on organic material [68]. By comparing CO_2_ data and performing a T-test, results imply that CTC had a significant effect on the biogas methane content (*p*-value = 0.05). At the same time, samples with tylosin only were not affected considerably. The reason could simply be that the CTC is active primarily against Gram-negative organisms by blending with the A location of the 30S subunit of bacterial ribosomes. So, they prevent peptide growth and the protein synthesis effect, which finally leads to bacteria death [65,69]. Methanogen bacteria are Gram-negative bacteria [70]. Thus, at a certain level of CTC, significant biogas inhibition should be imposed to the bioreactor.

Values of VS in the digestate before and after VA addition are shown in Figure 5. In general, every treatment sample except for the medium tylosin concentration, showed an increase in the VS percentage after VA addition. The VS values agree with the slight fluctuations observed in Figure 2, Figure 3 and Figure 4, that the microbial communities were slightly affected by the VA addition and the biogas production was not halted. To recall, the initial manure VS loading was 2 g-VS/L/day and the sampling for VS were conducted between 6 and 10 days after the first injection and two days after the last injection of VA. Thus, it complies with reduction in methane production which was around 13–15% studied by Dreher et al. [66]. Angenent et al. [34] also reported a temporary decrease in VS removal which recovered quickly. The average VS level before and after VA addition is 0.56% and 0.64%; and VS removal is 1.44% and 1.36%, respectively.

### 4.2. LC-MS/MS Results and Adjustments

The plan was to try using the measured Enrofloxacin concentrations to calculate adjustment factors for the other VAs. With these factors, sample concentrations after the dilution and short-term loading in the lab digesters were recalculated, assuming there was no degradation or absorption. Should there be significant differences, these would then be caused by sampling error, degradation due to AD, or the error of the LC-MS/MS measurement, including the SPE. Table 6 shows all of the samples, their added VAs and a comparison between spiked concentration, detected concentration by LC-MS/MS and recalculated concentration using adjustment factors.

### 4.3. Relatively High Recovery of Enrofloxacin in the Water-Only Samples

Although the spiked Enrofloxacin in digestate and manure samples had a very low recovery rate (226 ppb to 1433 ppb vs. 4444 ppb spiked values) (Table 7), all but one water sample detected relatively higher Enrofloxacin concentrations (1667 ppb to 3197 ppb, Table 1). The Enrofloxacin concentration in the first water sample (236 ppb, samples #1) was only a fraction of water samples. The water samples were made with distilled water and VAs, no solid manure. Relatively higher recovery rates suggest that there is a systematic bias in measuring the Enrofloxacin in the samples that have solids (manure and digestate).

Therefore, when sample 13 was excluded, the average of the water sample group was 2481 ppb, while the digestate samples averaged 825 ppb. On the other hand, if we disregard the Enrofloxacin concentrations as an adjustment factor and just compare the LC-MS/MS values with our calculated concentrations (assuming no degradation), provides a better outcome. In this way, external standards are utilized to evaluate samples with only water and VAs, to monitor whether the removal of VAs is due to AD or not.

### 4.4. Very High Recovery of Tylosin in the Water-Only Samples

As shown in Table 8, concentrations for tylosin are very close to and sometimes higher than what we were expecting (LC-MS/MS measured 106 ppm, we expected 92 ppm for sample 2). Furthermore, the water samples that were not spiked with tylosin did yield very low tylosin concentrations (0.86 ppm and 1.79 ppm, samples 13 and 16).

It is a different case for CTC; yet the reasons for low CTC detection are still unknown. The trend for CTC concentration shows they are disappearing so fast, which may be due to its half-life degradation or anaerobic reactor removal; alternatively, this may simply be because the CTC we used was already degraded, see the discussion below.

### 4.5. Consistent and Proportional LC-MS/MS Tylosin Results in the Digestate Samples

Based on the tylosin results being more consistent than CTC and Enrofloxacin results, the LC-MS/MS results of the digestate samples were meticulously scrutinized. Even though the LC-MS/MS detected concentration values of tylosin that were lower than expected, they were consistently proportional to the concentrations. For example, the expected concentration of tylosin was 20, 40, and 60 ppm for samples 4, 5, and 6, with zero degradation assumption, the LC-MS/MS values were 0.5, 0.9, and 1.2 ppm. No significant correlation for CTC was found. Figure 6 shows the LC-MS/MS results (*Y*-axis, ppm) vs. spiked values (*X*-axis, ppm). The consistently lower measured concentrations in the digestate samples and the high recovery rates in the water samples suggest that there was significant tylosin degradation due to the AD process.

### 4.6. Consistent LC-MS/MS Tylosin Measurements in the Manure External Standards

The detected tylosin concentrations of the three external standard samples were similar and had low deviation, Table 9. For tylosin, the detected levels ranged from 39.8 to 44.3 ppm and averaged 41.5 ppm, while the expected concentration was 77.9 ppm. For CTC, the measured concentrations were again a small fraction of the expected level.

### 4.7. Relatively Higher Recovery of CTC in the External Standard Samples, and Two Additional CTC Standards

Studying the results of the three external standard samples showed that there was a relatively higher recovery rate for CTC. As an instance, compared with the concentration of 233.78 ppm, LC-MS/MS detected 14.96, 36.16 and 28.52 ppm. Compared with previous CTC samples, this group had a much higher recovery rate, which was fresh samples made with diluted manure and VAs. Besides, the external standards were prepared with commercial-grade antibiotics instead of analytical grade.

Table 10 shows results for freshly prepared samples with diluted manure and CTC antibiotic, at concentrations of 4ppm and 40 ppm, and the LC-MS/MS measured concentrations were 1.63 ppm and 11.7 ppm, respectively. The recovery rates of CTC were 29% and 41%. Since the samples were freshly prepared, the probability of degradation due to AD or half-life degradation was eliminated. Other possibilities are absorption to organic matter, and that inconsistent purity or degradation had already happened before application. For tylosin, LC-MS/MS detected higher concentrations (12 ppm and 67.6 ppm detected for 4 ppm and 40 ppm samples, respectively), which gives a detection rate of 300% and 169%.

### 4.8. Applying External Standard Correction Factors

Because the internal standards (Enrofloxacin) did not yield consistent measurement, the correction was made based on external standards instead. By applying the external standard adjustment factor, the VA concentrations were corrected accordingly. The adjustment factor was obtained from samples with manure and spiked antibiotics, without retention time for AD. In other words, we just spiked different concentrations of antibiotics in samples made with manure, then prepared those for LC-MS/MS, immediately. In this way, we may be able to track other important factors contributing to our results, such as absorption, ion suppression or enhancement and recovery rate. Figure 7 presents the measured and corrected VA concentrations.

By using the external standard correction factor instead of the Enrofloxacin correction factor, data are more consistent, especially for tylosin (less than 6% error). It suggests that Enrofloxacin failed to act as an ideal internal standard. The inconsistency could be due to Enrofloxacin binding to the abundant organic materials or the presence of Ca^2+^, Mg^2+^ Ions.

### 4.9. Degradation of CTC

For CTC, results showed a high degradation rate for both the samples in water and AD (Figure 8). For instance, almost all CTC injections with various concentrations have close to zero concentration. The low concentrations were measured for AD-treated samples, and also for CTC dissolved in water stored at room temperature and 40 °C, the temperature of the AD. In addition, the concentrations of the external standards were 234 ppm. The results suggest that the CTC degrade much faster than the tylosin, which might be due to the shorter half-life (8 days) as reported by the manufacturer. It is also possible that the CTC powder we used had already degraded. CTC concentration in external standard samples was reduced to 131, 320 and 251 ppm from its original 234 ppm. Because the purity correction was already applied and recovery rate adjustment was made, also only low CTC concentration was detected for the water and digestate samples, the CTC probably just degraded itself over a short time. A study by Winckler and Grafe [71] showed that tetracycline in liquid manure was degraded by 50% in 82 days. Arikan [72] reported a 75% reduction in CTC concentration with AD after 33 days, with a half-life of 18 days. Cheng et al. [73] reported a high affinity between tetracycline and solid manure during AD. For future research, additional testing to examine the possibility of the adsorption by the glass jar used in this study as the AD reactors should be conducted, since there are very few investigations on this subject.

### 4.10. Degradation of Tylosin

Tylosin degraded very well with the ASBR reactor working at 39 °C and loading with swine manure every two days. By comparing the degradation rate of tylosin in ASBRs with jars filled only with water, it shows that AD is effective in reducing tylosin (Figure 9). The degradation rate of tylosin in water averaged 33.5; however, the degradation was 100 percent with AD. A study by Kolz et al. [74] concluded no effective degradation for tylosin B and D in anaerobic conditions up to eight months. tylosin A was degraded under the aerobic conditions with a half-life of 2 to 40 days [5,75]. Stone et al. [59] also reported no significant degradation for tylosin.

### 4.11. Effects of Having Two Types of VA in the Digestion and Water

In these experiments, we planned to compare the effect of having two different antibiotics on the reduction efficiency of antibiotics. Identical injections and concentrations were applied in the mixture treatment, except for a series of samples, both antibiotics were spiked.

For tylosin, the results showed that there was little difference between samples. The tylosin removal was similar with or without CTC mixture, suggesting that the CTC addition had no adverse effect on tylosin degradation.

However, because CTC degradation was much faster than tylosin, and because the samples collected for CTC concentration measurements were not resolute enough (shorter time than the five-day sampling), the speed of the CTC degradation and effects of the tylosin addition could not be determined based on this dataset (Table 9).

### 4.12. Contrast Tylosin Reduction in Water and AD Reactors

Tylosin tartrate showed a relatively higher removal in AD treatment when compared to samples that were dissolved in water. Figure 10 depicts that the tylosin samples treated by AD for twelve days were removed entirely (100% removal). However, the removal of tylosin tartrate in water (dotted line) was 40% or less during the same period. The lower degradation in the water samples suggests that AD is effective in enhancing tylosin degradation in the animal manure, and that this could be the essential effect of the AD.

### 4.13. Bacterial Community Dynamics

Phyla *Firmicutes*, *Bacteroidetes, Proteobacteria,* and *Synergistetes* were dominant or co-dominant in bacteria. Different types of *Clostridia* consisted mostly of the *Firmicutes*. *Methanomicrobia* was the dominant *Archaea* among our samples.

The first group was treated with CTC and showed a slight fluctuation in *Archaea* abundance, 4.37%, 5.19%, and 5.06% related to low, medium, and high concentrations of CTC, respectively. As shown in the Table 11, *Firmicutes* almost remained constant, and Bacteriodetes increased from 14.03 to 15.30% and then decreased to 12.50% for low, medium and high concentrations of CTC, respectively. The same pattern occurred for Synergistetes, going up from 1.67 to 1.75% and then dropping to 1.44%, with the mentioned level of CTC concentrations. However, the reverse happened for Proteobacteria: the abundance level reduced from 2.46 to 1.35% and then rose to 1.87%.

The second group, consisting of four samples, all are added with the mixture of CTC and tylosin with low, medium and high concentrations. Two samples were taken from medium range concentration because the administration duration of CTC was 6 days and tylosin was 10 days. Therefore, samples were taken at the end of the administration of each VA, at day 6 and day 10. There is a reverse relation between *Archaea* abundance and VAs concentration as well as administration duration. *Archaea* level is dropping with a higher concentration of VAs and longer retention time. The effect of the administration period is stronger than the dosage on the *Archaea* population. *Firmicutes*, on the other hand, have increased from 74.9 to 79.70% by increasing the dosage of VAs from low to medium. The trend is not consistent with shifting from medium to high concentration of VA; it would decrease the abundance of *Firmicutes*. The administration period has the same effect, but not as much as dosage. Phylum Bacteroides population increased with the rise of VAs concentration and doubled with an increase in retention time, from 6 to 10 days with a medium level of VAs. Proteobacteria and Synergistetes abundance both have dropped by increasing dosage for low to medium, but the fall is drastic for Proteobacteria, changing from 7 to 1.8%. By the end of CTC administration, when adding tylosin, their abundance recovered slightly. Surprisingly, in high concentrations of CTC and tylosin, both of these bacteria showed growth in their population. The proteobacteria population is almost doubled by having high dosage if VAs instead of low dosage.

The last group, which is being medicated with tylosin only, *Archaea*, decreased slightly and then almost doubled when moving from low concentration to high. For Bacteroides and *Firmicutes*, it is exactly the reverse, with maximum abundance around medium range concentration and the nearly same number for low and high concentrations. For proteobacteria, results showed a sharp drop with shifting from low to medium concentration, 10.2% changed to 4.03%. However, by raising the dosage, the abundance of proteobacteria returned to 15.97%.

*Bacteroidia* are the major classes found within the phylum of *Bacteroidetes;* and are abundant in digesters that use cow manure as feedstock [76]. *Firmicutes* phylum is mostly syntrophic bacteria that can decompose a variety of fatty acids, and exists in both activated sludge systems and anaerobic digesters [77]. Within the species of *Firmicutes*, *Clostridia* is the dominant class. The predominance of *Clostridia* in the AD sludge was related to the comparably fast hydrolysis and VFA (volatile fatty acids) fermentation happening in the digesters [78].

Fatty acid-oxidizing bacteria, including Synergistales group which have syntrophs are connecting bonds of the chain between the primary fermenters and methanogens [79], are abundant in thermophilic digesters [80,81]. The presence of *Synergistetes* (syntrophic acetate oxidizers) might be an indicator of a decent acetotrophic activity in the bioreactor [82].

There are two major categories of methanogens; *acetoclastic* which consumes acetate to produce methane or *hydrogenotrophic* that are converting CO_2_ and H_2_ to methane.

The *acetoclastic methanogenesis* is linked with the *Methanosarcinales* and the *hydrogenotrophic methanogenesis* is linked with the *Methanomicrobiaceae* family. In the current study, the *hydrogenotrophic* pathway with *Methanosarcinaceae* was dominant. Kim et al. [42], Nogueira et al. [78] and Padmasiri et al. [83] also detected a dominant *Methanomicrobiales* order on AD.

### 4.14. Statistical Analysis of the Effect of Different Treatments on Samples

There is a notable difference between the treated and control group, which shows antibiotics had a significant influence on altering the bacterial community in our digesters. Figure 11 shows the samples arranged using the same two similarity measures used to generate the PCoA plots but in the form of a dendrogram. Bray–Curtis similarities and Jaccard similarities method are used for this comparison.

In both of the methods, the differences between all three treatment groups are modest and likely obscured by the variability introduced by the control samples.

Figure 12 shows a stacked bar chart at the best taxonomic resolution afforded by our primers. Again, the differences between the two datasets are stark, while the differences between treatment groups are more subtle (but present).

Figure 13 shows a heat map in which samples (columns) are ordered according to similarity using a hierarchical method (UPGMA) based on the 50 OTUs (operational taxonomic unit) (rows) with the lowest p values following serial ANOVA testing of all 629 OTUs. In short, it shows perfect discrimination between groups when the samples are clustered using only the 50 most variable OTUs. Taxonomic identity of the microbes is listed on the right-hand side of the heat map.

Figure 14 shows box plots representing the relative abundance of eight of the OTUs with the lowest p values. One can easily note the very clear pattern of microbes with sensitivity to one or the other drugs, with Proteiniphilum (lower left) being the anomaly. Some other microbes with low p values mostly had these types of patterns, either down in CTC and CTC + tylosin, or down in tylosin and CTC + tylosin.

Finally, Figure 15 is a random forest analysis looking for “biomarkers” of each treatment group. The greater the “MeanDecreaseAccuracy”, the better that OTU is as a biomarker of the rankings shown to the right of the Figure. For example, Methanoculleus is apparently an excellent predictor of these groups by having a high relative abundance in the CTC samples and low abundance in the CTC + tylosin samples. Likewise, the Anaerorhabdus furcosa group and Flexilinea sp. can be found more on samples with high tylosin concentration and low CTC. Ruminiclostridium sp. can be abundant in conditions with high CTC levels and low tylosin.

## 5. Conclusions

The results show that for both CTC and tylosin with maximum concentration added of 298 and 263 ppm, respectively, a negligible inhibitory effect on ASBR performance was observed. No harmful effect on the microbial community, pH or alkalinity was observed; however, microbial diversity was decreased. Efficient tylosin removal with AD occurred (removal was 100%, while removal in distilled water-filled reactors was around 40% or less), though, it cannot be proven for CTC. In addition, no difference was detected for using a mixture of tylosin and CTC, compared to the solo use of each. More research must be carried out on testing different VAs to discover the efficiency of AD reactors for VA removal. Besides, amplicon sequencing performed on each group by using the 50 most variable operational taxonomic units (OTUs)s and perfect discriminations were detected between groups. The effect of administration period and dosage of VAs on Phyla *Firmicutes* Proteobacteria, Synergistetes and Phylum Bacteroides was investigated. OTU’s alteration is used to detect biomarkers. These biomarkers’ abundance can be employed to predict the sample’s contamination with these antibiotics.

## Figures and Tables

**Figure 1 bioengineering-07-00123-f001:**
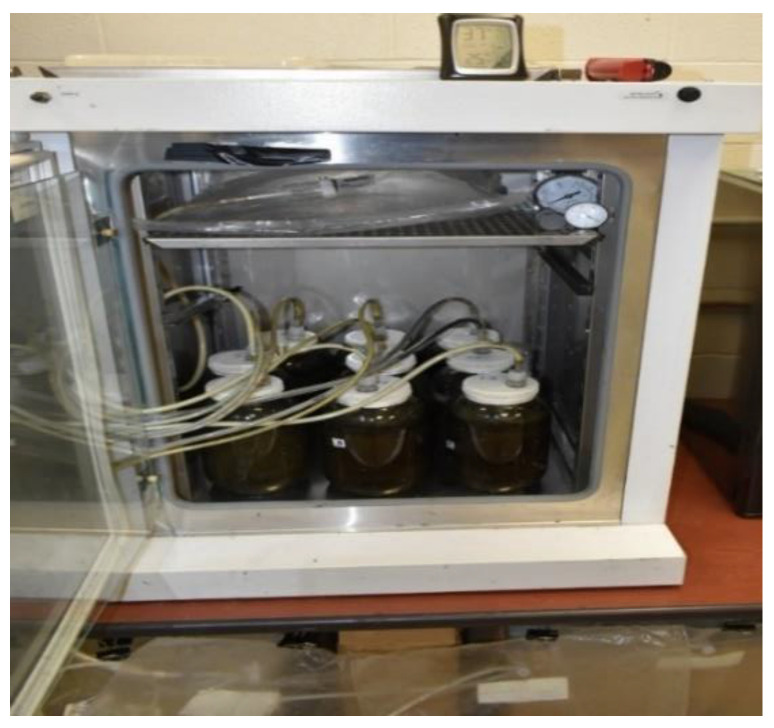
The scheme of the reactors, tubes transferring biogas and the incubator.

**Figure 2 bioengineering-07-00123-f002:**
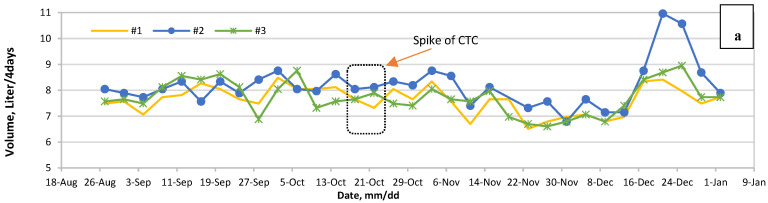
Biogas production in 2017–2018, (**a**) #1 low, #2 medium and #3 high concentration, before and after adding CTC, (**b**) #4 low, #5 medium and #6 high concentration, before and after adding both CTC and tylosin, and (**c**) #7 w, #8 medium and #9 high concentration, before and after adding tylosin and the dotted box shows the administration period of the VAs.

**Figure 3 bioengineering-07-00123-f003:**
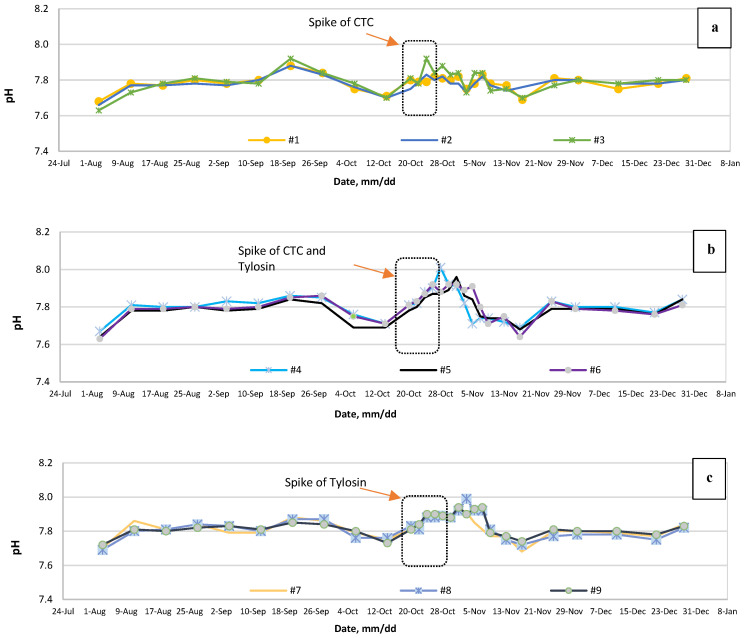
Digestate pH levels for (**a**) before and after CTC spike, (**b**) before and after CTC plus tylosin spike, and (**c**) before and after tylosin spike and the dotted box shows the administration period of the VAs.

**Figure 4 bioengineering-07-00123-f004:**
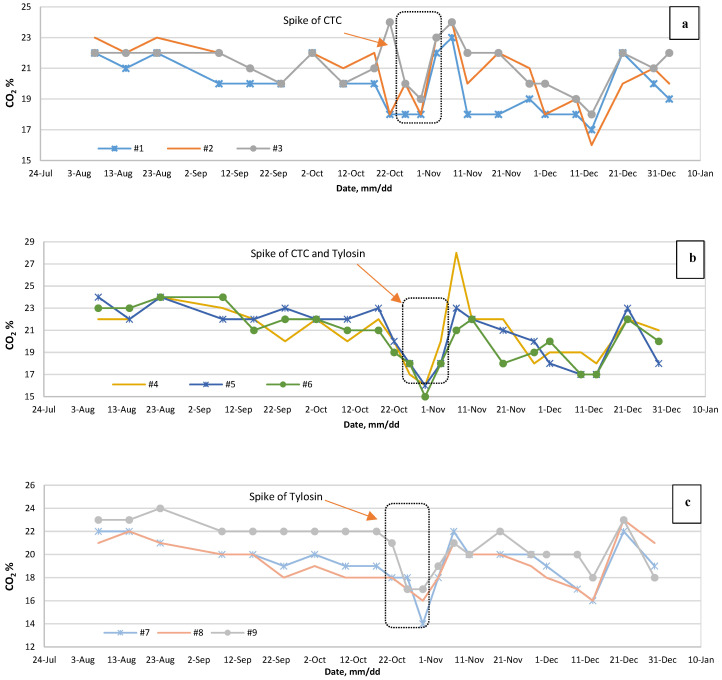
The concentrations of CO_2_ (**a**) before and after CTC spike, (**b**) before and after CTC plus tylosin spike, and (**c**) before and after tylosin spike and the dotted box shows the administration period of the VAs.

**Figure 5 bioengineering-07-00123-f005:**
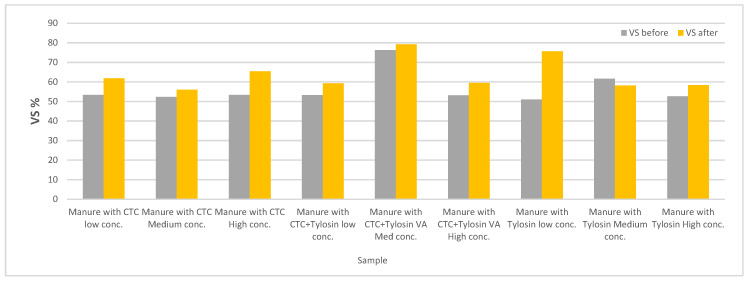
Digestate volatile solid (VS) concentrations before and after VA additions.

**Figure 6 bioengineering-07-00123-f006:**
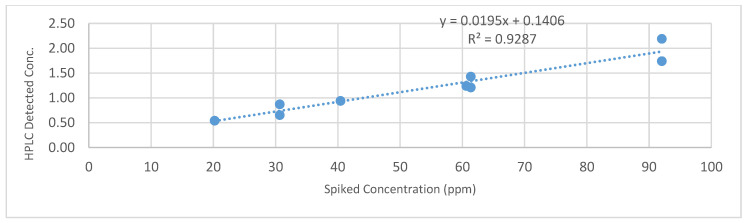
Correlations of LC-MS/MS measured and spiked tylosin concentrations.

**Figure 7 bioengineering-07-00123-f007:**
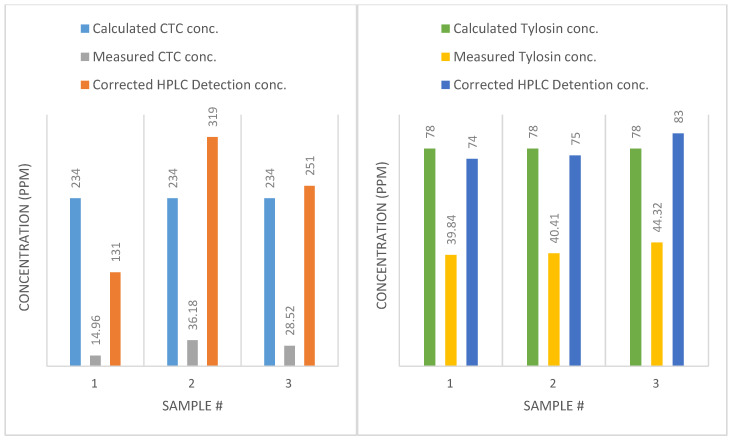
**Left**: concentrations of CTC, comparing calculated, measured, and corrected concentrations. **Right**: concentrations of tylosin, comparing calculated, measured, and corrected concentrations.

**Figure 8 bioengineering-07-00123-f008:**
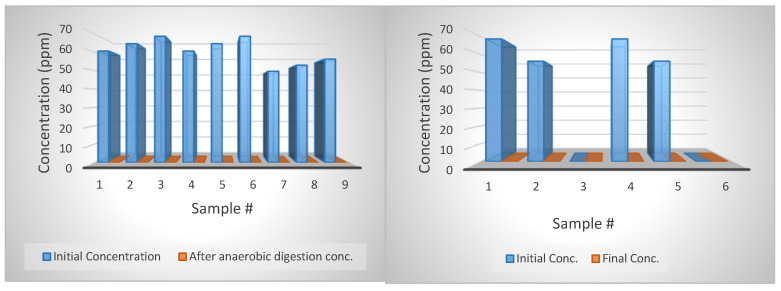
**Left**: CTC concentration change with anaerobic digestion. **Right**: CTC concentration change with reactors filled with diluted water.

**Figure 9 bioengineering-07-00123-f009:**
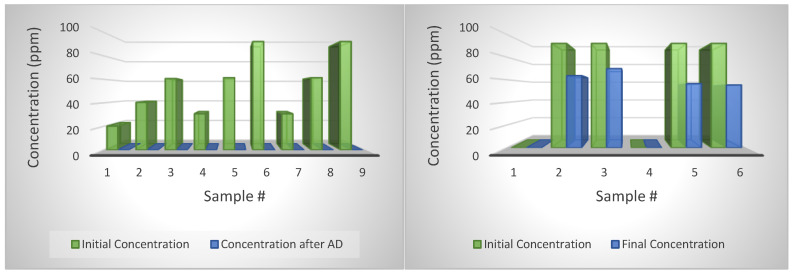
**Left**: Tylosin concentration change with anaerobic digestion. **Right**: Tylosin concentration change with reactors filled with diluted water.

**Figure 10 bioengineering-07-00123-f010:**
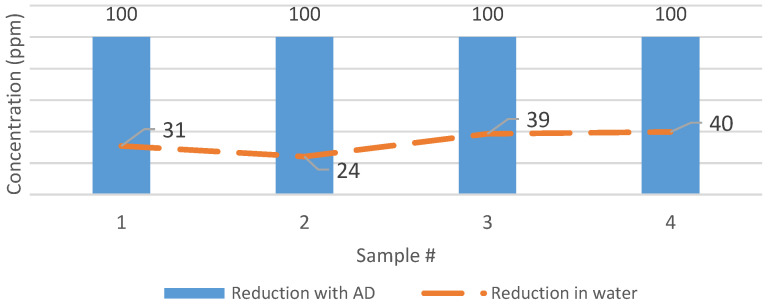
The removal of tylosin in water and the anaerobic digestion (AD) reactor.

**Figure 11 bioengineering-07-00123-f011:**
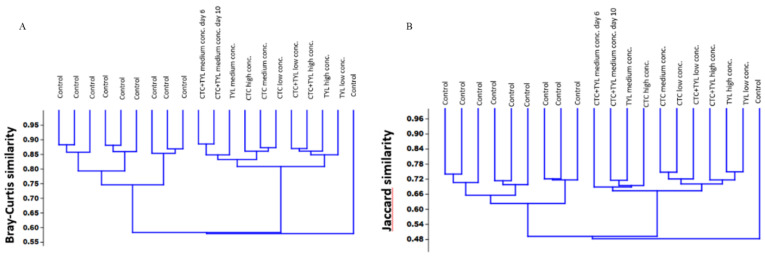
Dendrogram of bacterial community based on (**A**) Bray–Curtis similarities and (**B**) Jaccard similarities.

**Figure 12 bioengineering-07-00123-f012:**
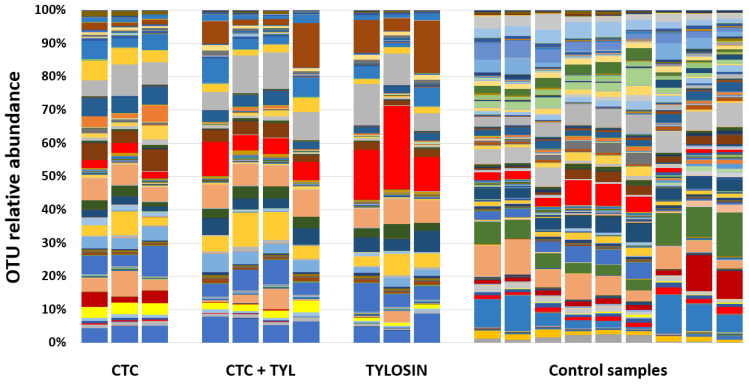
The stacked bar chart at the best taxonomic resolution.

**Figure 13 bioengineering-07-00123-f013:**
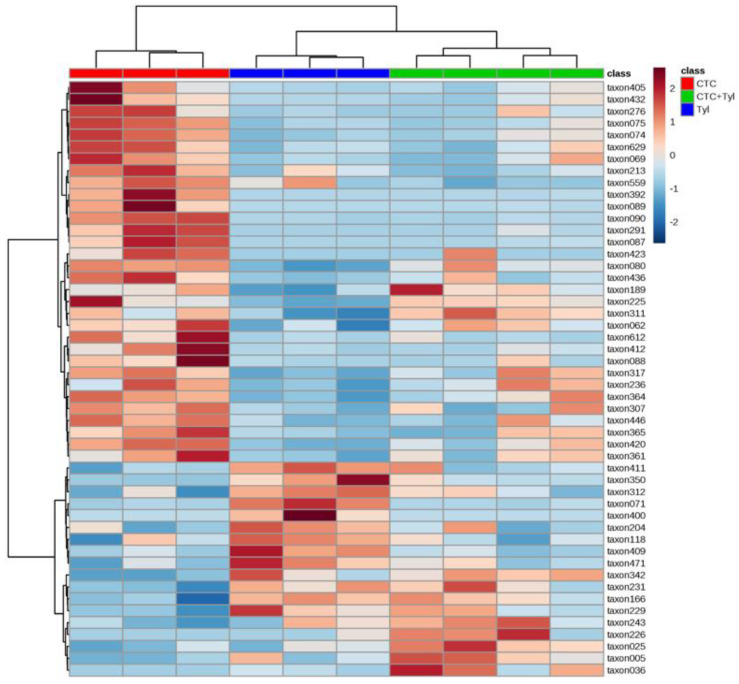
Heat map of treated samples using a hierarchical method (UPGMA) based on the 50 operational taxonomic units (OTUs) (rows) with the lowest p values.

**Figure 14 bioengineering-07-00123-f014:**
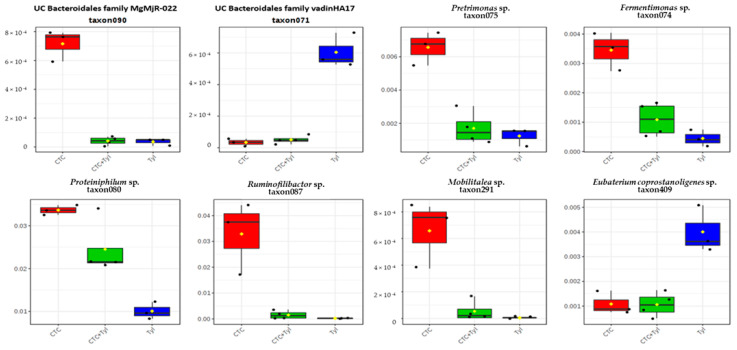
Box plots representing the relative abundance of 8 of the OTUs with the lowest *p* values.

**Figure 15 bioengineering-07-00123-f015:**
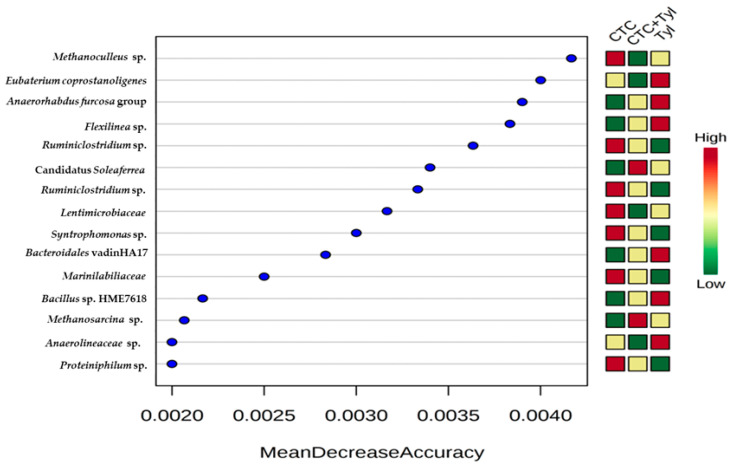
Random forest analysis of biomarkers based on MeanDecreaseAccuracy.

**Table 1 bioengineering-07-00123-t001:** Chlortetracycline (CTC) and tylosin water solubility.

Reference	CTC	Tylosin
**Manufacture Info.**	264 mg/L	528 mg/L
**Merck Index**	500 mg/L	6000 mg/L
**Sigma**	8.6 mg/mL	50 mg/mL

**Table 2 bioengineering-07-00123-t002:** Summary of antibiotics’ prescription and concentrations added every two days.

Antibiotic	Dosage	Manure Per Day (L/day)	Consumption	Treatment Duration	Excretion Level	Concentration (Lower Band) (mg/L)	Concentration (Upper Band) (mg/L)	Avg. (mg/L)
**Tylosin**	66 mg/liter	4.28	Drinking 11.35–18.93 L/day	3–10 days	50 to 90%	87.67	263	175.34
**CTC**	22 mg/kg.-body weight, day	4.28	Average Pig weight = 68 kg	3–5 days	75%	263	298	280.54

**Table 3 bioengineering-07-00123-t003:** Veterinary antibiotics (VAs) concentration spiked in each reactor.

Jar #	CTC Concentration (ppm)	Tylosin Concentration (ppm)
**1**	263	0
**2**	280.54	0
**3**	298	0
**4**	263	87.67
**5**	280.54	175.34
**6**	298	263
**7**	0	87.67
**8**	0	175.34
**9**	0	263

**Table 4 bioengineering-07-00123-t004:** Comparative table of methane content between data collected by Bacharach Fyrite Classic Combustion Analyzers and gas chromatography.

	Retention Time (ms)			Detected Volume		Bacharach Fyrite Classic Results	GC Results	
Sample #				CH_4_ (µL)	CO_2_ (µL)	CO_2_ (%)	CH_4_/100%	CO_2_/100%
1	120,043	384,994	283,623	117.17	57.50	28	56.51	27.73
2	67,869.1	441,945	300,395	134.25	60.85	28	62.86	28.49
3	68,710.1	455,328	311,024	138.27	62.98	28	62.86	28.63
4	131,615	411,986	302,588	125.26	61.29	26	56.33	27.56
5	131,197	421,019	309,690	127.97	62.71	26	56.52	27.70

**Table 5 bioengineering-07-00123-t005:** The ionization mode, retention times and optimized precursor/product ions for analysis of the VAs by the developed LC-MS/MS method.

	Chemical	Ionization Mode	Retention Time	Precursor/Product Ions
**1**	Ceftiofur (Excenel)	ESI+	7.04	523.8/210
**2**	Penicillin G Potassium salt	ESI+	7.32	335/160
**3**	Carbodox	ESI+	6.27	263/90
**4**	Chlortetracycline hydrochloride	ESI+	7.57	479/444
**5**	Tiamulin (Denaguard)	ESI+	8.64	494.3/192.1
**6**	Tylosin	ESI+	8.63	917/174
		ESI+	8.63	917/772
**7**	Enrofloxacin-d5	ESI+	6.98	365/321

**Table 6 bioengineering-07-00123-t006:** Samples, their content and comparison between their calculated, detected by LC-MS/MS and adjusted concentrations.

Sample #	Sample Type	Calculated CTC Conc. (ppm)	CTC Detected Conc. with LC-MS/MS (ppm)	Calculated Tylosin Conc. (ppm)	Tylosin Detected Conc. with LC-MS/MS (ppm)	CTC Recovery and Purity Adj. (ppm)	Tylosin Recovery and Purity Adj. (ppm)	Enrofloxacin Conc. (ppb)
**1**	Digestate with CTC low conc.	61	0	0	0	−1	0	226
**2**	Digestate with CTC Medium conc.	65	0	0	0	−2	0	523
**3**	Digestate with CTC High conc.	69	0	0	0	−1	0	1377
**4**	Digestate with Mixture VA low conc.	61	0	20	1	0	−6	1069
**5**	Digestate with Mixture VA Med conc.	65	0	40	1	−2	−5	940
**6**	Digestate with Mixture VA High conc.	69	0	61	1	−2	−4	568
**7**	Digestate with Mixture VA low conc.	50	0	31	1	−1	−5	1434
**8**	Digestate with Mixture VA Med conc.	53	0	61	1	−1	−4	600
**9**	Digestate with Mixture VA High conc.	56	0	92	2	−2	−2	834
**10**	Digestate with tylosin low conc.	0	0	31	1	−2	−5	881
**11**	Digestate with tylosin Medium conc.	0	0	61	1	−2	−4	1186
**12**	Digestate with tylosin High conc.	0	0	92	2	−2	−3	891
**13**	VA in water, Heat treated	69	0	0	1	−1	−1	236
**14**	VA in water, Heat treated	56	0	92	106	−1	64	3197
**15**	VA in water, Heat treated	0	0	92	116	−1	70	1668
**16**	VA in water, Room Temp.	69	0	0	2	−1	−1	1775
**17**	VA in water, Room Temp.	56	0	92	95	−1	57	2809
**18**	VA in water, Room Temp.	0	0	92	93	−1	55	2957
**19**	Diluted Manure without VA	0	0	0	1	−2	−4	839
**20**	Diluted Manure without VA	0	0	0	1	−2	−4	777
**21**	Diluted Manure without VA	0	0	0	1	−2	−5	613
**22**	Diluted Manure with CTC + tylosin	234	15	78	40	131	74	675
**23**	Diluted Manure with CTC + tylosin	234	36	78	40	319	75	874
**24**	Diluted Manure with CTC + tylosin	234	29	78	44	251	83	552

**Table 7 bioengineering-07-00123-t007:** Concentrations of VAs were detected in the water samples.

Sample #	Sample Type	Compound	Detected CTC Conc.(ppb)	Detected Tylosin Conc. (ppb)	Enrofloxacin Conc. (ppb)
**13**	VA in water, Heat treated	CTC	24	859	236
**14**	VA in water, Heat treated	CTC + tylosin	15	106,243	3197
**15**	VA in water, Heat treated	Tylosin	4	116,324	1668
**16**	VA in water, Room Temp.	CTC	17	1787	1775
**17**	VA in water, Room Temp.	CTC + tylosin	9	94,837	2809
**18**	VA in water, Room Temp.	Tylosin	17	93,002	2957

**Table 8 bioengineering-07-00123-t008:** Comparison of calculated VAs concentrations and LC-MS/MS detected levels.

Sample #	Sample Type	Compound	Calculated CTC Conc. (ppm)	CTC Detected Conc. with LC-MS/MS (ppm)	Calculated Tylosin Conc. (ppm)	Tylosin Detected Conc. with LC-MS/MS (ppm)
**13**	VA in water, Heat treated	CTC	68.74	0.02	0.00	0.86
**14**	VA in water, Heat treated	CTC + tylosin	56.27	0.01	92.07	106.24
**15**	VA in water, Heat treated	Tylosin	0.00	0.00	92.07	116.32
**16**	VA in water, Room Temp.	CTC	68.74	0.02	0.00	1.79
**17**	VA in water, Room Temp.	CTC + tylosin	56.27	0.01	92.07	94.84
**18**	VA in water, Room Temp.	Tylosin	0.00	0.02	92.07	93.00

**Table 9 bioengineering-07-00123-t009:** Concentrations of VAs were detected in the manure external standard samples.

Sample #	Sample Type	Compound	Calculated CTC Conc. (ppm)	CTC Detected Conc. with LC-MS/MS (ppm)	Calculated Tylosin Conc. (ppm)	Tylosin Detected Conc. with LC-MS/MS (ppm)
**1**	Diluted Manure with VAs	CTC + tylosin low Conc.	233.78	14.96	77.93	39.84
**2**	Diluted Manure with VAs	CTC + tylosin low Conc.	233.78	36.18	77.93	40.41
**3**	Diluted Manure with VAs	CTC + tylosin low Conc.	233.78	28.52	77.93	44.32

**Table 10 bioengineering-07-00123-t010:** Concentrations of measured calibration standards.

Spiked Conc. (ppm)	LC-MS/MS Detected Conc. (ppm)	
	CTC	Tylosin
**0**	0.03	1.26
**4**	1.63	12.17
**40**	11.69	67.62

**Table 11 bioengineering-07-00123-t011:** Percentages of different microorganisms with various treatment plans.

	Total	CTC Low	CTC Med	CTC High	CTC + Tyl Low	CTC + Tyl Med	CTC + Tyl Med	CTC + Tyl High	Tyl Low	Tyl Med	Tyl High
D_0__Archaea; D_1__Euryarchaeota	5.90%	4.40%	5.20%	5.10%	7.80%	7.40%	5.20%	6.30%	4.80%	3.70%	8.70%
D_0__Bacteria; D_1__Actinobacteria	0.10%	0.10%	0.00%	0.10%	0.10%	0.10%	0.10%	0.10%	0.10%	0.10%	0.10%
D_0__Bacteria; D_1__Atribacteria	1.00%	1.00%	1.10%	1.50%	0.90%	1.10%	0.90%	1.30%	0.90%	0.50%	0.70%
D_0__Bacteria; D_1__Bacteroidetes	7.70%	14.00%	15.30%	12.50%	4.00%	5.90%	10.20%	5.50%	2.20%	5.30%	2.50%
D_0__Bacteria; D_1__Chloroflexi	0.40%	0.50%	.40%	0.30%	0.40%	0.30%	0.30%	0.70%	0.80%	0.30%	0.50%
D_0__Bacteria; D_1__Cloacimonetes	0.10%	0.10%	0.30%	0.10%	0.00%	0.10%	0.20%	0.00%	0.00%	0.10%	0.00%
D_0__Bacteria; D_1__Firmicutes	74.70%	72.80%	71.90%	75.30%	74.90%	79.70%	77.90%	68.10%	76.50%	82.70%	66.70%
D_0__Bacteria; D_1__Kiritimatiellaeota	1.00%	0.80%	0.80%	0.50%	1.60%	0.90%	0.70%	0.90%	1.50%	1.00%	1.60%
D_0__Bacteria; D_1__Planctomycetes	0.30%	0.40%	0.30%	0.30%	0.30%	0.10%	0.20%	0.10%	0.30%	0.20%	0.40%
D_0__Bacteria; D_1__Proteobacteria	6.00%	2.50%	1.40%	1.90%	7.00%	1.80%	1.90%	13.40%	10.20%	4.00%	16.00%
D_0__Bacteria; D_1__Synergistetes	2.00%	1.70%	1.70%	1.40%	2.60%	1.80%	1.30%	3.40%	2.10%	1.50%	2.60%
No blast hit; Other	0.60%	1.50%	1.50%	0.80%	0.20%	0.50%	0.90%	0.10%	0.40%	0.40%	0.10%

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
