# Peer review of "Degradation of Veterinary Antibiotics in Swine Manure via Anaerobic Digestion"

_bioengineering, 2020, doi:10.3390/bioengineering7040123_

Round 1
Reviewer 1 Report
In this manuscript, the author investigated the transmission of veterinary antibiotic (tylosin and ) and its impact on anaerobic digestion. The main deficiency is the overall structure is disordered while the author gave abundant information and details from each part. Currently, degradation of antibiotics during anaerobic digestion and the potential impact on microbial communities have been intensively investigated with various focuses, for instance temperature, reactors, and more. The author did not state the novelty of this work, why this work should be published and what is the main difference than others. In addition, I suggest re-organizing the entire manuscript, for instance, there are two parts from introduction, while the second part, methane production, can be merged into the first part and abundant information should be deleted.
Some general comments:
- The abstract should be revised and write it in short. There are unnecessary details included, for instance ‘Line 20-21, Biogas produced by each AD was collected using 10-L Tedlar bags for volume measurement, which was conducted every four days.’ It good enough to have that in M&M part.
Experimental set-up-.
- In general, the experimental set-up is in a good manner. However, in details, it is still questionable. As I understood, the reactor was operated continuously until the antibiotics was spiked. Afterwards, antibiotics spiked manure was fed into the reactor following the administration period (with equal volume discharged); which moved to sequence batch test (SBR)? (Under continuous configuration, the concentrations of antibiotics will not reach to stable without operation of minimum one HRT, if the degradation rate is unchanged). That is why I thought it might not be a good idea to consider the administration period only.
In addition, Tylosin is one of the most used antibiotics that has been intensively investigated and reported regarding the its degradation during anaerobic digestion. Back to this administration time, the first sampling day for tylosin was day 10, which is much longer than what was typically reported; tylosin is the most fast-degraded antibiotics which can be eliminated completely even within 24 hours (depending on applied concentration). Please try to give more details about this. Why is this administration time still matter if the residual tylosin is excreted from pig?
R&D
- In total, there are 20 figures included in the manuscript, which is much more than regular figure numbers in a manuscript. I suggest to reduce the numbers of figures (as supplementary file or just describe in the text if it is not very informative )(Figure 2-4, 5-7, 8-10 can be merged as figure 2, 3, 4 (a, b, c) for instance). The X-axis of those figures were marked as the operated date, I suggest to use test days instead as it is tightly overlapped and unnecessary.
- It seems the measured concentration is much low than the calculated or corrected. What is the reason and how could you ensure the corrected value is reliable ? Was there any suspected or unexpected loss during extraction ?
- The language, especially from R&D part, should be revised and improved.
Some details:
Some details are included as well.
Line 13, VA (veterinary antibiotic) should be rewrote as ‘veterinary antibiotic (VA)’
Line 49, ‘daily use of it rocketed during the last two decades. (FDA summary report, 2016).’ Full stop after decades should be deleted.
Line 51, VA raised by 36.8 average of ? % or times ?
Line 51, what is ‘weak retention’ ? unstable during AD ?
Line 100, ‘depending on organic material. (Graaf and Fendler, 2010).’ Full stop should not be used between the text and followed citation.
In the introduction part, the author should include the state of the art, the deficiency of the former studies, and the novelty of current study. It is confusing to divide introduction to two parts (introduction and methane production). I suggest to merge those two parts together and re-structure the introduction.
Line 151, I suggest to list the geographical information of the swine farm.
Line 158, Volatile suspended solids, or volatile solids ?
Line 163-164. A literature was cited after the TS and VS data. I don’t think it is necessary.
Line 181, were be emptied should be were emptied.
Line 196-213, these parts are redundant, especially when used in M&M part.
I suggest keeping SI unit only, the aim is to publish in an international journal anyway.
Figure 12/13, please do not use 3-D column graph.
Line 590-591, it is true, five days for 1st sample is too long to monitor the degradation.
(Currently, it is not quite easy to give detailed comment/suggestion for R&D part. It would be easier if this part can be improved).
Reviewer 2 Report
The reviewed article discusses and investigates the very current issue of the degradation of antibiotics present in swine manure subjected to AD process. The presence of antibiotics in wastewater, sewage sludge or agricultural by-products can cause numerous problems and interfere with the efficiency of their processing. Therefore, the research issue undertaken by the authors is particularly important in both scientific and technological terms.
The research experiment was well planned and conducted. The authors described the methods and research tools used in detail. The results of the study are extensively discussed and graphically illustrated, which allowed the authors to draw the correct conclusions. The presented results and conclusions will be of interest to readers, as they relate not only to the effectiveness of the degradation of antibiotics in the AD process , but also to the impact of antibiotics on microbes present in the process and the course of the AD process itself. The number of drawings and tables contained in the article is impressive, although that can sometimes overwhelm the reader of the reviewed text.
Reviewer 3 Report
- In abstract, on lines 19-21 - you wrote: "produced biogas and digester pH were measured weekly. Biogas produced by each AD was collected using 10-L Tedlar bags for volume measurement, which 20 was conducted every four days - please clarity how often the biogas was measured
- In methods section - please add reference when you explain why you used these concentrations of VAs, please extend information about concentration these VAa in the feedstock for animal and VAs impact on environment.
- you froze a raw material - Was examined the effect of freezing on the efficiency of methane fermentation? (freezing is a method of pretreatment)
- reference at line 164 - please explain why you used this reference in paragraph about characteristic of inoculum.
- Please explain why you decide that- reactors were fed every two days, not every day.
- What about composition of biogas? Especially methane content
- units - please use SI units not e.g. lb. or gal
- were repetitions used in the studies? - there is no information in the text on this topic
- table 4 – unit for volume
- lack of reference in the section describe sample analysis (e.eg. pH, GC for composition of biogas - methane concentration )
- Figures 2-4 - please add information which means: # 1 … etc.- under the title of the
- Please compare the process efficiency (biogas/ methane yield, VS removal – please calculate these parameter) with the typical values for AD of swine manure
- Results and discussion – lack reference and consequently discussion about degradation of Veterinary Antibiotics; you show only results with any comments
- sequencing batch reactors (SBR)? – you meant ASBR, right?
- please correct name of parameter – not removal rate, because it is the amount of material removed per time unit, but e.g. TCC removal etc.
- where are the results for alkalinity?
- Why was VFA/alkalinity ratios not analyzed? (one of the basic indicators of process stability)
I understand that the most important issue in the article was to determine the effectiveness of the removal of selected antibiotics. Nevertheless, the article should contain basic information on the efficiency of the anaerobic digestion, such as: VS removal, methane/biogas yield, composition of biogas. For this reason, please calculate mentioned parameter.
Round 2
Reviewer 1 Report
I strongly suggest the author to revise the introduction and shape the structure. It is very unusual to have 15-16 individual paragraphs in introduction.
In addition, anaerobic digestion is the process itself while anaerobic digester is the device carrying on the process. Therefore, it is necessary to use AD as abbreviation for anaerobic digestion only while use full name or digester for another.
Author Response
Dear Reviewers
The introduction has been reorganized, and unnecessary part are removed. We tried to shorten this section, the introduction section has six paragraphs now. thanks.
All the anaerobic digesters that were mentioned with the acronym ADs, have changed to their full name now. The abbreviation “AD” is used only for anaerobic digestion in this manuscript. Thanks.